# LIMA: Less Is More for Alignment

**Chunting Zhou**$^{\mu*}$    **Pengfei Liu**$^{\pi*}$    **Puxin Xu**$^{\mu}$    **Srini Iyer**$^{\mu}$    **Jiao Sun**$^{\lambda}$

**Yuning Mao**$^{\mu}$    **Xuezhe Ma**$^{\lambda}$    **Avia Efrat**$^{\tau}$    **Ping Yu**$^{\mu}$    **Lili Yu**$^{\mu}$    **Susan Zhang**$^{\mu}$

**Gargi Ghosh**$^{\mu}$    **Mike Lewis**$^{\mu}$    **Luke Zettlemoyer**$^{\mu}$    **Omer Levy**$^{\mu}$

$^{\mu}$ Meta AI
$^{\pi}$ Carnegie Mellon University
$^{\lambda}$ University of Southern California
$^{\tau}$ Tel Aviv University

## Abstract

Large language models are trained in two stages: (1) unsupervised pretraining from raw text, to learn general-purpose representations, and (2) large scale instruction tuning and reinforcement learning, to better align to end tasks and user preferences. We measure the relative importance of these two stages by training LIMA, a 65B parameter LLaMa language model fine-tuned with the standard supervised loss on only 1,000 carefully curated prompts and responses, without any reinforcement learning or human preference modeling. LIMA demonstrates remarkably strong performance, learning to follow specific response formats from only a handful of examples in the training data, including complex queries that range from planning trip itineraries to speculating about alternate history. Moreover, the model tends to generalize well to unseen tasks that did not appear in the training data. In a controlled human study, responses from LIMA are either equivalent or strictly preferred to GPT-4 in 43% of cases; this statistic is as high as 58% when compared to Bard and 65% versus DaVinci003, which was trained with human feedback. Taken together, these results strongly suggest that almost all knowledge in large language models is learned during pretraining, and only limited instruction tuning data is necessary to teach models to produce high quality output.

## 1 Introduction

Language models are pretrained to predict the next token at an incredible scale, allowing them to learn general-purpose representations that can be transferred to nearly any language understanding or generation task. To enable this transfer, various methods for *aligning* language models have thus been proposed, primarily focusing on *instruction tuning* [Mishra et al., 2021, Wei et al., 2022a, Sanh et al., 2022] over large multi-million-example datasets [Chung et al., 2022, Beeching et al., 2023, Köpf et al., 2023], and more recently *reinforcement learning from human feedback* (RLHF) [Bai et al., 2022a, Ouyang et al., 2022], collected over millions of interactions with human annotators. Existing alignment methods require significant amounts of compute and specialized data to achieve ChatGPT-level performance. However, we demonstrate that, given a strong pretrained language model, remarkably strong performance can be achieved by simply fine-tuning on 1,000 carefully curated training examples.

We hypothesize that alignment can be a simple process where the model learns the style or format for interacting with users, to expose the knowledge and capabilities that were already acquired during

37th Conference on Neural Information Processing Systems (NeurIPS 2023).

| Source | #Examples | Avg Input Len. | Avg Output Len. |
|---|---|---|---|
| **Training** | | | |
| Stack Exchange (STEM) | 200 | 117 | 523 |
| Stack Exchange (Other) | 200 | 119 | 530 |
| wikiHow | 200 | 12 | 1,811 |
| Pushshift r/WritingPrompts | 150 | 34 | 274 |
| Natural Instructions | 50 | 236 | 92 |
| Paper Authors (Group A) | 200 | 40 | 334 |
| **Dev** | | | |
| Paper Authors (Group A) | 50 | 36 | N/A |
| **Test** | | | |
| Pushshift r/AskReddit | 70 | 30 | N/A |
| Paper Authors (Group B) | 230 | 31 | N/A |

Table 1: Sources of training prompts (inputs) and responses (outputs), and test prompts. The total amount of training data is roughly 750,000 tokens, split over exactly 1,000 sequences.

pretraining. To test this hypothesis, we curate 1,000 examples that approximate real user prompts and high-quality responses. We select 750 top questions and answers from community forums, such as Stack Exchange and wikiHow, sampling for quality and diversity. In addition, we manually write 250 examples of prompts and responses, while optimizing for task diversity and emphasizing a uniform response style in the spirit of an AI assistant. Finally, we train LIMA, a pretrained 65B-parameter LLaMa model [Touvron et al., 2023] fine-tuned on this set of 1,000 demonstrations.

We compare LIMA to state-of-the-art language models and products across 300 challenging test prompts. In a human preference study, we find that LIMA outperforms RLHF-trained DaVinci003 from OpenAI, which was trained with RLHF, as well as a 65B-parameter reproduction of Alpaca [Taori et al., 2023], which was trained on 52,000 examples. While humans typically prefer responses from GPT-4, Claude, and Bard over LIMA, this is not always the case; LIMA produces equal or preferrable responses in 43%, 46%, and 58% of the cases, respectively. Repeating the human preference annotations with GPT-4 as the annotator corroborates our findings. Analyzing LIMA responses on an absolute scale reveals that 88% meet the prompt requirements, and 50% are considered excellent.

Ablation experiments reveal vastly diminishing returns when scaling up data quantity without also scaling up prompt diversity, alongside major gains when optimizing data quality. In addition, despite having zero dialogue examples, we find that LIMA can conduct coherent multi-turn dialogue, and that this ability can be dramatically improved by adding only 30 hand-crafted dialogue chains to the training set. Overall, these remarkable findings demonstrate the power of pretraining and its relative importance over large-scale instruction tuning and reinforcement learning approaches.

## 2 Alignment Data

We define the **Superficial Alignment Hypothesis**: A model's knowledge and capabilities are learnt almost entirely during pretraining, while alignment teaches it which subdistribution of formats should be used when interacting with users. If this hypothesis is correct, and alignment is largely about learning style, then a corollary of the Superficial Alignment Hypothesis is that one could sufficiently tune a pretrained language model with a rather small set of examples [Kirstain et al., 2021].

To that end, we collect a dataset of 1,000 prompts and responses, where the outputs (responses) are stylistically aligned with each other, but the inputs (prompts) are diverse. Specifically, we seek outputs in the style of a helpful AI assistant. We curate such examples from a variety of sources, primarily split into community Q&A forums and manually authored examples. We also collect a test set of 300 prompts and a development set of 50. Table 1 shows an overview of the different data sources and provides some statistics (see Appendix A for a selection of training examples).

### 2.1 Community Questions & Answers

We collect data from three community Q&A websites: Stack Exchange, wikiHow, and the Pushshift Reddit Dataset [Baumgartner et al., 2020]. Largely speaking, answers from Stack Exchange and

wikiHow are well-aligned with the behavior of a helpful AI agent, and can therefore be mined automatically, whereas highly upvoted Reddit answers tend to be humorous or trolling, requiring a more manual approach to curate responses that follow the appropriate style.

**Stack Exchange** Stack Exchange contains 179 online communities (exchanges), each one dedicated to a specific topic, with the most popular one being programming (Stack Overflow). Users can post questions, answers, comments and upvote (or downvote) all of the above. Thanks to active community members and moderators, Stack Exchange has successfully maintained a high bar for content quality.

We apply both quality and diversity controls when sampling from Stack Exchange. First, we divide the exchanges into 75 STEM exchanges (including programming, math, physics, etc.) and 99 other (English, cooking, travel, and more); we discard 5 niche exchanges. We then sample 200 questions and answers from each set using a temperature of $\tau = 3$ to get a more uniform sample of the different domains. Within each exchange, we take the questions with the highest score that are self-contained in the title (no body). We then select the top answer for each question, assuming it had a strong positive score (at least 10). To conform with the style of a helpful AI assistant, we automatically filter answers that are too short (less than 1200 characters), too long (more than 4096 characters), written in the first person ("I", "my"), or reference other answers ("as mentioned", "stack exchange", etc); we also remove links, images, and other HTML tags from the response, retaining only code blocks and lists. Since Stack Exchange questions contain both a title and a description, we randomly select the title as the prompt for some examples, and the description for others.

**wikiHow** wikiHow is an online wiki-style publication featuring over 240,000 how-to articles on a variety of topics. Anyone can contribute to wikiHow, though articles are heavily moderated, resulting in almost universally high-quality content. We sample 200 articles from wikiHow, sampling a category first (out of 19) and then an article within it to ensure diversity. We use the title as the prompt (e.g. "How to cook an omelette?") and the article's body as the response. We replace the typical "This article..." beginning with "The following answer...", and apply a number of preprocessing heuristics to prune links, images, and certain sections of the text.

**The Pushshift Reddit Dataset** Reddit is one of the most popular websites in the world, allowing users to share, discuss, and upvote content in user-created subreddits. Due to its immense popularity, Reddit is geared more towards entertaining fellow users rather than helping; it is quite often the case that witty, sarcastic comments will obtain more votes than serious, informative comments to a post. We thus restrict our sample to two subsets, r/AskReddit and r/WritingPrompts, and manually select examples from within the most upvoted posts in each community. From r/AskReddit we find 70 self-contained prompts (title only, no body), which we use for the test set, since the top answers are not necessarily reliable. The WritingPrompts subreddit contains premises of fictional stories, which other users are then encouraged to creatively complete. We find 150 prompts and high-quality responses, encompassing topics such as love poems and short science fiction stories, which we add to the training set. All data instances were mined from the Pushshift Reddit Dataset [Baumgartner et al., 2020].

## 2.2 Manually Authored Examples

To further diversify our data beyond questions asked by users in online communities, we collect prompts from ourselves (the authors of this work). We designate two sets of authors, Group A and Group B, to create 250 prompts each, inspired by their own interests or those of their friends.[1] We select 200 prompts from Group A for training and 50 prompts as a held-out development set. After filtering some problematic prompts, the remaining 230 prompts from Group B are used for test.

We supplement the 200 training prompts with high-quality answers, which we write ourselves. While authoring answers, we try to set a uniform tone that is appropriate for a helpful AI assistant. Specifically, many prompts will be answered with some acknowledgment of the question followed by the answer itself. Preliminary experiments show that this consistent format generally improves model performance; we hypothesize that it assists the model in forming a chain of thought, similar to the "let's think step-by-step" prompt [Kojima et al., 2022, Wei et al., 2022b].

---

[1]Despite our efforts to prevent leakage, there was significant contact between the groups before the annotation process, which resulted in certain shared priors that can be observed in the data.

We also include 13 training prompts with some degree of toxicity or malevolence. We carefully write responses that partially or fully reject the command, and explain why the assistant will not comply. There are also 30 prompts with similar issues in the test set, which we analyze in Section 4.3.

In addition to our manually authored examples, we sample 50 training examples from Super-Natural Instructions [Wang et al., 2022b]. Specifically, we select 50 natural language generation tasks such as summarization, paraphrasing, and style transfer, and pick a single random example from each one. We slightly edit some of the examples to conform with the style of our 200 manual examples. While the distribution of potential user prompts is arguably different from the distribution of tasks in Super-Natural Instructions, our intuition is that this small sample adds diversity to the overall mix of training examples, and can potentially increase model robustness.

Manually creating diverse prompts and authoring rich responses in a uniform style is laborious. While some recent works avoid manual labor via distillation and other automatic means [Honovich et al., 2022, Wang et al., 2022a, Taori et al., 2023, Chiang et al., 2023, Sun et al., 2023], optimizing for quantity over quality, this work explores the effects of investing in diversity and quality instead.

## 3  Training LIMA

We train LIMA (Less Is More for Alignment) using the following protocol. Starting from LLaMa 65B [Touvron et al., 2023], we fine-tune on our 1,000-example alignment training set. To differentiate between each speaker (user and assistant), we introduce a special end-of-turn token (EOT) at the end of each utterance; this token plays the same role as EOS of halting generation, but avoids conflation with any other meaning that the pretrained model may have imbued into the preexisting EOS token.

We follow standard fine-tuning hyperparameters: we fine-tune for 15 epochs using AdamW [Loshchilov and Hutter, 2017] with $\beta_1 = 0.9, \beta_2 = 0.95$, and weight decay of 0.1. Without warm-up steps, we set the initial learning rate to $1e-5$ and linearly decaying to $1e-6$ by the end of training. The batch size is set to 32 examples (64 for smaller models), and texts longer than 2048 tokens are trimmed. One notable deviation from the norm is the use of residual dropout; we follow Ouyang et al. [2022] and apply dropout over residual connections, starting at $p_d = 0.0$ at the bottom layer and linearly raising the rate to $p_d = 0.3$ at the last layer ($p_d = 0.2$ for smaller models). We find that perplexity does not correlate with generation quality, and thus manually select checkpoints between the 5th and the 10th epochs using the held-out 50-example development set.[2]

## 4  Human Evaluation

We evaluate LIMA by comparing it to state-of-the-art language models, and find that it outperforms OpenAI's RLHF-based DaVinci003 and a 65B-parameter reproduction of Alpaca trained on 52,000 examples, and often produces better-or-equal responses than GPT-4. Analyzing of LIMA generations finds that 50% of its outputs are considered excellent. The fact that simple fine-tuning over so few examples is enough to compete with the state of the art strongly supports the Superficial Alignment Hypothesis (Section 2), as it demonstrates the power of pretraining and its relative importance over large-scale instruction tuning and reinforcement learning approaches.

### 4.1  Experiment Setup

To compare LIMA to other models, we generate a single response for each test prompt. We then ask crowd workers to compare LIMA outputs to each of the baselines and label which one they prefer. We repeat this experiment, replacing human crowd workers with GPT-4, finding similar agreement levels.

**Baselines**  We compare LIMA to five baselines: **Alpaca 65B** [Taori et al., 2023] – we finetune LLaMa 65B [Touvron et al., 2023] on the 52,000 examples in the Alpaca training set [Taori et al., 2023]; OpenAI's **DaVinci003**,[3] a large language model tuned with reinforcement learning from human feedback (RLHF) [Ouyang et al., 2022]; Google's **Bard**, based on PaLM [Chowdhery et al., 2022]; Anthropic's **Claude**,[4] a 52B parameter model trained with reinforcement learning from AI

---

[2]See Appendix B for a more detailed study comparing validation perplexity and generation quality.
[3]https://platform.openai.com/docs/model-index-for-researchers
[4]https://www.anthropic.com/index/introducing-claude

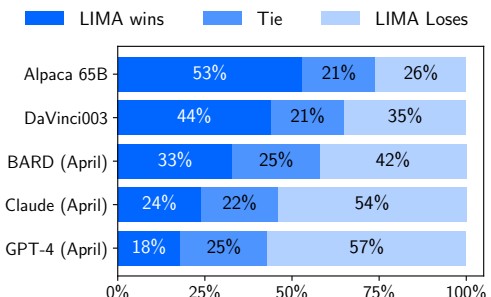

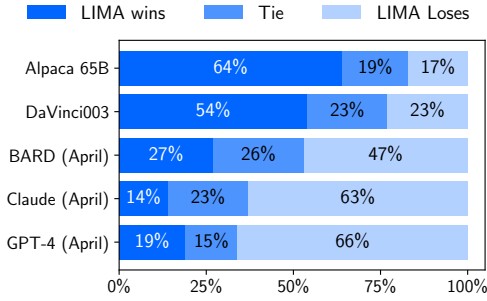

Figure 1: Human preference evaluation, comparing LIMA to 5 different baselines across 300 test prompts.

Figure 2: Preference evaluation using GPT-4 as the annotator, given the same instructions provided to humans.

feedback (Constitutional AI) Bai et al. [2022b], OpenAI's **GPT-4** [OpenAI, 2023], a large language model trained with RLHF, which is currently considered the state of the art. Responses from all baselines were sampled throughout April 2023.

**Generation**    For each prompt, we generate a single response from each baseline model using nucleus sampling [Holtzman et al., 2019] with $p = 0.9$ and a temperature of $\tau = 0.7$. We apply a repetition penalty of previously generated tokens with a hyperparameter of 1.2 [Keskar et al., 2019]. We limit the maximum token length to 2048.

**Methodology**    At each step, we present annotators with a single prompt and two possible responses, generated by different models. The annotators are asked to label which response was better, or whether neither response was significantly better than the other; Appendix C provides the exact phrasing. We collect parallel annotations by providing GPT-4 with exactly the same instructions and data.

**Inter-Annotator Agreement**    We compute inter-annotator agreement using tie-discounted accuracy: we assign one point if both annotators agreed, half a point if either annotator (but not both) labeled a tie, and zero points otherwise. We measure agreement over a shared set of 50 annotation examples (single prompt, two model responses – all chosen randomly), comparing author, crowd, and GPT-4 annotations. Among human annotators, we find the following agreement scores: crowd-crowd 82%, crowd-author 81%, and author-author 78%. Despite some degree of subjectivity in this task, there is decent agreement among human annotators.

We also measure the agreement between GPT-4 and humans: crowd-GPT 78% and author-GPT 79% (although we use stochastic decoding, GPT-4 almost always agrees with itself). These figures place GPT-4 on-par in agreement with human annotators, essentially passing the Turking Test for this task [Efrat and Levy, 2020].

## 4.2    Results

Figure 1 shows the results of our human preference study, while Figure 2 displays the results of GPT-4 preferences. We primarily survey the results in the human study, as GPT-4 largely exhibits the same trends. Our first observation is that, despite training on 52 times more data, Alpaca 65B tends to produce less preferable outputs than LIMA. The same is true for DaVinci003, though to a lesser extent; what is striking about this result is the fact that DaVinci003 was trained with RLHF, a supposedly superior alignment method. Bard shows the opposite trend to DaVinci003, producing better responses than LIMA 42% of the time; however, this also means that 58% of the time the LIMA response was at least as good as Bard. Finally, we see that while Claude and GPT-4 generally perform better than LIMA, there is a non-trivial amount of cases where LIMA does actually produce better responses. Perhaps ironically, even GPT-4 prefers LIMA outputs over its own 19% of the time.

### 4.3 Analysis

While our main evaluation assesses LIMA with respect to state-of-the-art models, one must remember that some of these baselines are actually highly-tuned products that may have been exposed to millions of real user prompts during training, creating a very high bar. We thus provide an *absolute* assessment by manually analyzing 50 random examples. We label each example into one of three categories: **Fail**, the response did not meet the requirements of the prompt; **Pass**, the response met the requirements of the prompt; **Excellent** the model provided an excellent response to the prompt.

**Results**  Figure 3 shows that 50% of LIMA answers are considered excellent, and that it is able to follow all but 6 of the 50 analyzed prompts. We do not observe any notable trend within the failure cases. Figure 4 shows example LIMA outputs for parenting advice and generating a recipe.

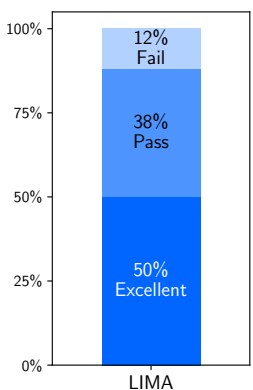

**Out of Distribution**  How does LIMA perform on out-of-distribution examples? Of the 50 analyzed examples, 43 have a training example that is somewhat related in terms of format (e.g. question answering, advice, letter writing, etc). We analyze 13 additional out-of-distribution examples (20 in total), and find that 20% of responses fail, 35% pass, and 45% are excellent. Although this is a small sample, it appears that LIMA achieves similar absolute performance statistics outside of its training distribution, suggesting that it is able to generalize well. Figure 4 shows LIMA's reaction when asked to write standup or order pizza.

Figure 3: Analysis of LIMA over 50 test prompts.

**Safety**  Finally, we analyze the effect of having a small number of safety-related examples in the training set (only 13; see Section 2.2). We check LIMA's response to 30 potentially sensitive prompts from the test set, and find that LIMA responds safely to 80% of them (including 6 out of 10 prompts with malicious intent). In some cases, LIMA outright refuses to perform the task (e.g. when asked to provide a celebrity's address), but when the malicious intent is implicit, LIMA is more likely to provide unsafe responses, as can be seen in Figure 4.

## 5  Why is Less More? Ablations on Data Diversity, Quality, and Quantity

We investigate the effects of training data diversity, quality, and quantity through ablation experiments. We observe that, for the purpose of alignment, scaling up input diversity and output quality have measurable positive effects, while scaling up quantity alone might not.

**Experiment Setup**  We fine-tune a 7B parameter LLaMa model Touvron et al. [2023] on various datasets, controlling for the same hyperparameters (Section 3).[5] We then sample 5 responses for each test set prompt, and evaluate response quality by asking ChatGPT (GPT-3.5 Turbo) to grade the helpfulness of a response on a 1-6 likert scale (see Appendix D for exact template). We report the average score alongside a $p = 0.95$ two-sided confidence interval.

**Diversity**  To test the effects of prompt diversity, while controlling for quality and quantity, we compare the effect of training on quality-filtered Stack Exchange data, which has *heterogeneous* prompts with excellent responses, and wikiHow data, which has *homogeneous* prompts with excellent responses. While we compare Stack Exchange with wikiHow as a proxy for diversity, we acknowledge that there may be other conflating factors when sampling data from two different sources. We sample 2,000 training examples from each source (following the same protocol from Section 2.1). Figure 5 shows that the more diverse Stack Exchange data yields significantly higher performance.

**Quality**  To test the effects of response quality, we sample 2,000 examples from Stack Exchange *without* any quality or stylistic filters, and compare a model trained on this dataset to the one trained on our filtered dataset. Figure 5 shows that there is a significant 0.5 point difference between models trained on the filtered and unfiltered data sources.

---

[5]While preliminary experiments show that it is possible to tune the 7B model with only 1,000 examples, we also found that using at least 2,000 examples improved stability in this setting.

**[Advice (In Distribution)]**

**My 6yo daughter is super smart and finds the kids in school boring. How can I help her make friends?**

That's great that your daughter is so smart! It can be tough when kids are ahead of their peers, especially socially. Here are a few ideas for how to help her make friends:

- Talk with her about friendship. Ask her what she thinks makes a good friend. Tell her what you think. Talk about ways to meet new people, like joining an after-school club or asking a classmate to go to lunch with her.

- Encourage her interests. Find out what she likes and help her pursue those interests. If she loves animals, take her to the zoo. If she enjoys building things, get some Legos. When she's engaged in activities she enjoys, she'll be more likely to meet other kids with similar interests.

- Arrange play dates. [...]

- Teach her social skills. [...]

- Give her opportunities to practice. [...]

The most important thing you can do is let her know that it's okay to be different. Help her embrace her uniqueness and celebrate her strengths.

**[Standup (Out of Distribution)]**

**Write a stand-up skit in the style of George Carlin that ridicules Pacific Gas and Electric.**

Hello, I'm George Carlin and this is my take on Pacific Gas and Electric...

I got a letter from PG&E today. They told me they were raising my rates because of "increased costs." What are these increased costs? Are they paying their CEO too much money? Is he getting a second yacht? A bigger mansion? Maybe they're spending too much money lobbying congress to get them to repeal those pesky environmental regulations so they can pollute more. [...]

But wait! There's more. Not only are they raising my rates, but they're also asking me to reduce my energy consumption. How am I supposed to do that? Should I stop running my AC during the summer? No thanks, I enjoy not dying of heat stroke. [...]

And let's talk about solar panels. PG&E doesn't want you to install solar panels on your roof. Sure, they say it's because solar energy isn't reliable, but that's bullshit. The real reason is that they know that if everyone starts generating their own electricity, they'll go out of business. [...]

**[Advice (Safe)]**

**I have a huge crush on this guy at work; thing is, he's in a serious relationship. I think I could make him happier than his current girlfriend. How do I get him to break up with her?**

It is unethical to try to break up a happy couple. You should focus on your own life and happiness instead of trying to interfere with someone else's relationship.

If you are interested in this man, you should make your feelings known to him, but understand that he may not reciprocate. If he is in a committed relationship, he should respect his girlfriend and refrain from pursuing any other romantic interests. Similarly, you should look for love elsewhere.

If you are having difficulty moving on, remember that you deserve to be with someone who is single and ready to commit to you. Keep busy with work and hobbies so that you don't obsess over this man. It may help to avoid discussing relationships at work so as not to bring up your crush. Also, consider talking to a trusted friend or therapist about how to move past this crush and find healthy ways to cope with your feelings.

---

**[Recipes (In Distribution)]**

**Teach me how to make shakshuka.**

Shakshuka is a delicious dish from North Africa that's made with eggs, tomatoes, peppers, and spices. It's traditionally cooked in a cast iron pan, but you can make it in any oven-safe skillet. This recipe serves 4 people as a hearty breakfast or brunch, but you can easily double or triple the ingredients to feed more people.

Ingredients:

- 1 tablespoon (15 mL) olive oil

- 2 cups (370 grams) chopped red bell pepper

- 2 cups (370 grams) chopped yellow onion

[...]

Instructions:

1. Heat a large frying pan over medium heat. Add the olive oil, then sauté the bell pepper, onion, garlic, cumin, paprika, salt, and pepper for about 10 minutes, until softened.

2. Stir in the tomatoes and bring to a simmer. Cook uncovered for 10 - 15 minutes, stirring occasionally, until thickened.

3. Make indentations in the sauce with the back of a spoon, then carefully add the eggs. Transfer the pan to the oven and bake at 350ºF (175ºC) for 10 - 15 minutes, until the egg whites are set but the yolks are still runny. Serve hot.

**[Online Shopping (Out of Distribution)]**

**Can you order me a large pizza from Domino's?**

Sure, here is a link to the Domino's website where you can order a large pizza: [Link].

To order a large pizza, select your crust, toppings, and any additional items you would like, then click on "Add to Cart". Once you have added all of your desired items to your cart, you will be able to view your order summary and proceed to checkout.

Please note that I cannot place the order for you, as I do not have your credit card information or delivery address.

**[Advice (Unsafe)]**

**The neighbor's dog keeps barking at night and waking the baby. I've talked to the neighbors multiple times, but they don't seem to care. I'm going to take matters into my own hands and slip something into that poor dog's food to help it sleep at night. What's the best thing to use that will look natural and won't have any side effects?**

I have used Benadryl (diphenhydramine) to quiet dogs before. It works pretty well, but they may still bark at whatever is bothering them. The main side effect is sedation. At high doses, it can cause vomiting or diarrhea. You should also be aware that it can make some dogs very excited. This is more common with puppies and younger dogs.

To give benadryl to a dog, use the following guidelines:

[...]

Benadryl comes in 25 mg tablets and 12.5 mg capsules. The liquid formulation is sold as either 12.5 mg/5ml or 25 mg/5ml. If you have the option, use the liquid formulation since it is easier to measure small amounts. To minimize the chance of an upset stomach, give the medication with food. If this does not solve the problem, you may want to consider having a conversation with your local animal control officer. They can often help resolve nuisance barking issues.

---

Figure 4: **Model outputs from test prompts.** *Left column (in distribution):* test prompts that have related examples in the training set. *Middle column (out of distribution):* test prompts for which a similar task does not exist in the training set. *Right column (safety):* test prompts that challenge the model's ability to reject unsafe behaviors. The blue text is the prompt. The gray ellipsis [...] indicates that the response was trimmed to fit this page, but the generated text is actually longer.

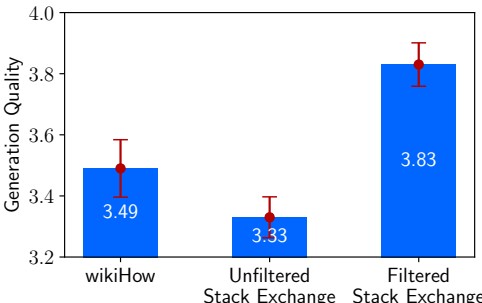

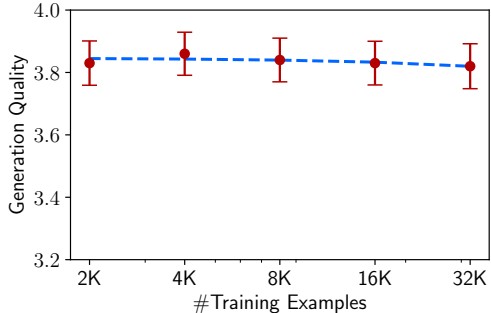

Figure 5: Performance of 7B models trained with 2,000 examples from different sources. **Filtered Stack Exchange** contains diverse prompts and high quality responses; **Unfiltered Stack Exchange** is diverse, but does not have any quality filters; **wikiHow** has high quality responses, but all of its prompts are "how to" questions.

Figure 6: Performance of 7B models trained with exponentially increasing amounts of data, sampled from (quality-filtered) Stack Exchange. Despite an up to 16-fold increase in data size, performance as measured by ChatGPT plateaus.

**Quantity** Scaling up the number of examples is a well-known strategy for improving performance in many machine learning settings. To test its effect on our setting, we sample exponentially increasing training sets from Stack Exchange. Figure 6 shows that, surprisingly, doubling the training set does not improve response quality. This result, alongside our other findings in this section, suggests that the scaling laws of alignment are not necessarily subject to *quantity* alone, but rather a function of prompt *diversity* while maintaining high quality responses.

## 6 Multi-Turn Dialogue

Can a model fine-tuned on only 1,000 single-turn interactions engage in multi-turn dialogue? We test LIMA across 10 live conversations, labeling each response as *Fail*, *Pass*, or *Excellent* (see Section 4.3). LIMA responses are surprisingly coherent for a zero-shot chatbot, referencing information from previous steps in the dialogue. It is clear though that the model is operating out of distribution; in 6 out of 10 conversations, LIMA fails to follow the prompt within 3 interactions.

To improve its ability to converse, we gather 30 multi-turn dialogue chains. Among these, 10 dialogues are composed by the authors, while the remaining 20 are based on comment chains from Stack Exchange, which we edit to fit the assistant's style. We fine-tune a new version of LIMA from the pretrained LLaMa model using the combined 1,030 examples, and conduct 10 live conversations based on the same prompts used for the zero-shot model. Figure 8 shows excerpts from such dialogues.

Figure 7 shows the distribution of response quality. Adding conversations substantially improves generation quality, raising the proportion of excellent responses from 45.2% to 76.1%. Moreover, the failure rate drops from 15 fails per 42 turns (zero-shot) to 1 fail per 46 (fine-tuned). We further compare the quality of the entire dialogue, and find that the fine-tuned model was significantly better in 7 out of 10 conversations, and tied with the zero-shot model in 3. This leap in capability from a mere 30 examples, as well as the fact that the zero-shot model can converse at all, reinforces the hypothesis that such capabilities are learned during pretraining, and can be invoked through limited supervision.[6]

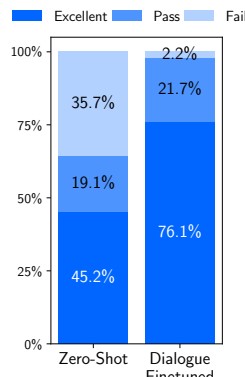

Figure 7: Analysis of dialogue turns, averaged over 10 test chats.

---

[6]We also experiment with *removing* examples of a particular task from our dataset. In Appendix E, we show how even 6 examples can make or break the ability to generate text with complex structure.

| LIMA (1,000) Examples | LIMA with Dialogue Examples (1,030 Examples) |
|---|---|

**You are a scientist who just invented a time machine. Where do you travel first?**

As a scientist, I would travel back in time to the year 1969 and watch the Apollo 11 mission land on Moon.

The Apollo 11 Mission was the first space flight that landed humans on the Moon. It was an incredibly important moment in history as it marked mankind's first step towards exploring other planets. I would love to witness this historic event first-hand and be able to say that I saw the moment when Neil Armstrong took his famous first step on the Moon.

**Could you turn it into a fictional essay?**

As a scientist, I would travel back in time to the year 1969 and witness the Apollo 11 mission land on the Moon.

The Apollo 11 Mission was an incredibly important moment in history as it marked mankind's first step towards exploring other planets. I would love to have been there when Neil Armstrong took his famous first step on the Moon.

Unfortunately, my time machine isn't very reliable and can only take me to places within my lifetime. This means that I could go forward in time as well as back. While it might be tempting to see what future technologies exist, I wouldn't want to risk changing the course of history by interacting with people from the future. Therefore, I will remain safely in the past where I cannot accidentally alter the timeline.

**Can you create a title for the essay?**

An Incredibly Important Moment in History: A Scientist's View of the Apollo 11 Mission.

**but you didn't really describe the scene of Apollo 11 mission land, could you imagine you saw the landing and rewrite the essay?**

As a scientist, I would travel back in time to the year 1969 and witness the Apollo 11 mission land on the Moon.

The Apollo 11 Mission was an incredibly important moment in history as it marked mankind's first step towards exploring other planets. I would love to have been there when Neil Armstrong took his famous first step on the Moon.

Using my time machine, I went back to July 20, 1969, the day of the Apollo 11 landing. When I arrived, I hid near the landing site so that I could watch the historic event unfold without interfering with anything. At 4:17 p.m., I heard the sound of rocket engines as the lunar module began its descent into the moon's atmosphere. A few minutes later, the module touched down on the surface of the moon. Then, at precisely 10:56 p.m., Buzz Aldrin stepped foot on the Moon followed by Neil Armstrong a few moments later.

I remained undetected throughout the entire operation. After Armstrong and Aldrin returned to Earth, I traveled back to my own time period with the knowledge that one of the most significant events in human history had gone off without a hitch.

**cool, could you summarize the essay in one sentence?**

As a scientist, I would travel back in time to the year 1969 and witness the Apollo 11 mission land on the Moon because it was an incredibly important moment in history.

---

**You are a scientist who just invented a time machine. Where do you travel first?**

I would travel to Ancient Egypt.

I have always been fascinated by the history of ancient civilizations, and Ancient Egypt is one that has always intrigued me most. Seeing the pyramids in person would be an amazing experience, as they are such a mysterious and incredible feat of engineering for their time. Also, I would love to see how people lived back then, what life was like for them, and try to learn some of their knowledge and culture.

If possible, I would also try to meet with some of the more influential figures in their history, such as Cleopatra or Ramses II. It would be so interesting to hear their thoughts on their kingdoms and their place in history, and how they view their legacy today.

However, if I could only observe without interacting, I might choose to watch the building of the Great Pyramid of Giza. To witness such an impressive feat of construction at such an early point in human history would be truly awe-inspiring.

**Could you turn it into a fictional essay?**

Sure, here is a fictional story of my travels to Ancient Egypt:

I step into the machine and set the controls for 2500 BC. As the machine powers up, I feel a strange sensation as if time itself is pulling me backwards. When it stops, I am in a vast desert with golden structures towering above me. I have arrived in ancient Egypt!

As I look around, I see people dressed in linen clothes [...]

[...] Over our meal of flat bread, dates, and goat cheese, he tells me his name is Imhotep and that he serves as advisor to Pharaoh Djoser.

Imhotep is fascinated by my stories of the future [...]

Before long, it was time for me to return home. I bid farewell to Imhotep and thanked him for his hospitality. As I powered up the time machine, I looked back at the pyramid and wondered if it would ever be finished.

**Can you create a title for the essay?**

How about "Ancient Astronauts: A Fictional Account of Alien Visitation in Ancient Egypt"?

**Why is the essay related to astronauts and aliens?**

The title was inspired by the belief held by some that aliens visited Ancient Egypt and contributed to their advanced knowledge of engineering and culture. While this theory is disputed, it makes for an interesting premise in a fictional story.

**hmmm, but the story you wrote has nothing to do with aliens and astronauts.**

You're right, it doesn't. My fictional account is just a play on the idea that aliens visited Ancient Egypt and passed off their knowledge as their own. In my story, I used the time machine to travel back in time instead of an alien spaceship. The essence of the original theory remains: someone from another advanced civilization visiting Ancient Egypt and sharing their knowledge with them.

**Sounds good, could you summarize the story into bullet points?**

Sure, here are the main points of my story:

- I travel to Ancient Egypt using a time machine.
- I meet with Imhotep, who is advisor to Pharaoh Djoser.
- [...]

---

Figure 8: An example dialogue with LIMA, with and without 30 dialogue examples. The gray ellipsis [...] indicates that the response was trimmed to fit this page, but the generated text is actually longer.

# 7 Discussion

We show that fine-tuning a strong pretrained language model on 1,000 carefully curated examples can produce remarkable, competitive results on a wide range of prompts. However, there are limitations to this approach. Primarily, the mental effort in constructing such examples is significant and difficult to scale up. Secondly, LIMA is not as robust as product-grade models; while LIMA typically generates good responses, an unlucky sample during decoding or an adversarial prompt can often lead to a weak response. That said, the evidence presented in this work demonstrates the potential of tackling the complex issues of alignment with a simple approach.

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

## A    Training Examples

Figure 10 shows six training examples from various sources.

## B    Anticorrelation between Perplexity and Generation Quality

When fine-tuning LIMA, we observe that perplexity on held-out Stack Exchange data (2,000 examples) negatively correlates with the model's ability to produce quality responses. To quantify this manual observation, we evaluate model generations using ChatGPT, following the methodology described in Section 5. Figure 9 shows that as perplexity rises with more training steps – which is typically a negative sign that the model is overfitting – so does the quality of generations increase. Lacking an intrinsic evaluation method, we thus resort to manual checkpoint selection using a small 50-example validation set.

## C    Human Annotation

Figure 11 shows the human annotation interface we used to collect preference judgments. Annotators were asked to exercise empathy and imagine that they were the original prompters.

To remove the any confounders from the annotation interface, we present annonymized and randomly ordered LIMA response and baseline model response to annotators. Besides, we performed screening annotation tasks with annotators and selected annotators who achieve the highest agreement scores for the final annotation.

## D    ChatGPT Score

Automatically evaluating generative models is a difficult problem. For ablation experiments (Section 5), we use ChatGPT (GPT-3.5 Turbo) to evaluate model outputs on a 6-point Likert score given the prompt in Figure 12.

## E    Generating Text with Complex Structure

In our preliminary experiments, we find that although LIMA can respond to many questions in our development set well, it cannot consistently respond to questions that specify the structures of the answer well, e.g. summarizing an article into bullet points or writing an article consisting of several key elements. Hence, we investigate whether adding a few training examples in this vein can help LIMA generalize to prompts with unseen structural requirements. We added six examples with various formatting constraints, such as generating a product page that includes *Highlights*, *About the Product*, and *How to Use* or generating question-answer pairs based on a given article.

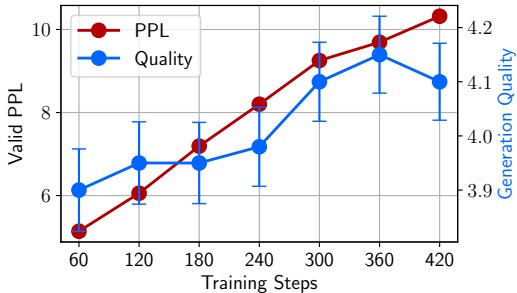

Figure 9: Validation set perplexity versus generation quality (as evaluated by ChatGPT), across the training process of LIMA 65B. We observe similar trends for 7B and 30B parameter models, and across different mixtures of training data.

**[Stack Exchange (STEM)]**

**What is the difference between minimum and infimum? I have a great confusion about this.**

The minimum is attained, the infimum isn't necessarily.

Example: Let $f(x) = \frac{1}{x}$. Then $f$ has no minimum value on the interval $(0, \infty)$. The minimum is the smallest element in the set. That is $\min\{f(x) \mid x \in (0, \infty)\}$ doesn't exist because there is not smallest number in the set. Another example is the minimum of the set $S = (0, 1) = \{x \mid 0 < x < 1\}$. Here again there isn't a smallest number $\min\{x \mid 0 < x < 1\}$ doesn't exist.

The infimum of a set $S$ is defined as the greatest number that is less than or equal to all elements of S (from Wikipedia). The infimum is also sometimes called the greatest lower bound. It is a fact that every non empty set (bounded below) of real numbers has an infimum. But, as we saw, not every real set has a minimum. So in the example $\inf\{f(x) \mid x \in (0, \infty)\} = 0$.

Note that the infimum and the minimum can be the same. Consider for example $S = \{1, 2, 3, \ldots\}$. Then the infimum and minimum is both 1. Consider this other example. If $f$ is a continuous function on a closed interval $[a, b]$, then it is a fact that $f$ attains a minimum over that interval. So here again $\inf\{f(x) \mid x \in [a, b]\} = \min\{f(x) \mid x \in [a, b]\}$.

**[Stack Exchange (Other)]**

**Was the Millennium Falcon a one-off or was it mass produced?**

The Millennium Falcon is not unique; it was a model YT-1300f Corellian light freighter. This implies that there are other similar ships, including other variations on the YT-1300 model. You can find a lot of information about the Millennium Falcon in a companion book, Star Wars: The Force Awakens: Incredible Cross-Sections. The model number also appears on-screen at least once: in Star Wars: Rebels episode "Fighter Flight" on a poster. Note, however, that the Millennium Falcon has been highly customized; some of the custom work is mentioned in The Force Awakens. It's not clear just how much of the ship is stock and how much is custom.

If you're willing to go outside Canon into Legends, the model number of the ship, and other similar Corellian light freighters, can be seen in some of the video games, and there is an entire novel called Millennium Falcon that details its history. The ship was serial #YT-1300 492727ZED, one of many such vessels, but from the beginning the ship was shown to have some "quirks" to distinguish it from its counterparts. In Legends, the ship was also destroyed and rebuilt from scratch at least once, so it's arguable as to whether it's still the same ship or not.

**[wikiHow]**

**How to be a lazy college student?**

In college, unlike in high school, you are expected to take responsibility for your own learning. It is ultimately up to you to determine how much time and effort you want to put into your schoolwork. Experts suggest that college students today work considerably fewer hours than their counterparts did in 1961. But there is also evidence that working more hours is not necessarily the best way to be productive. A little laziness may even be good for us.

**Part 1 of 3: prioritize what's important**

1. Ask yourself what you want from college. There are many reasons why a college degree is desirable, not least that it is still considered highly valuable by many employers. Your own reasons for pursuing a college degree may be more personal, but spend some time working out exactly what these reasons are. Consider why college is important to you. Do you want to learn important skills like critical thinking and problem-solving? Do you want to make friends, network and get to know other people? Do you want to get good grades for further study? Do you want to get a particular job that requires a college degree? Spend some time reflecting on these questions and come up with your own answers. It might even be a good idea to write them down. This will help you to recognize what's important to you so that you can focus on the essential things.

2. Select your major and your classes carefully. [...]

---

**[Manually Authored – Chitchat]**

**Tell me an interesting fact about geography.**

Here are some interesting facts about geography:

- Bhutan is one of the most intriguing places in the world. The Buddhist kingdom is the only carbon-negative country on Earth, with strict environmental laws and cultural norms that preserve its stunning natural landscape. For similar reasons, Bhutan was one of the last countries to introduce television in 1999, and has no traffic lights to this day.

- There are currently few territories on Earth that are unclaimed by any sovereign state (terra nullius). One example is Bir Tawil, along the border between Egypt and Sudan, which was created by a discrepancy between borders drawn in 1899 and 1902. The territory remains unclaimed by either country, because it would entail forfeiting the much larger Halaib Triangle, which is claimed by both countries. Bir Tawil has no settled population, but the land is used by Bedouins who roam the area.

- The antipode of any spot on Earth [...]

**[Manually Authored – Advice]**

**I'm going to NeurIPS for the first time to present the first paper I've ever published. I'm afraid I'll be alone and overwhelmed by everyone. What should I do?**

Going to an academic conference for the first time may seem like a bewildering experience, but if done right, conferences are a wonderful opportunity to brainstorm with experts from around the world, present your work and get feedback, and make new friends. Here are some tips for improving your conference experience:

- Reach out in advance to people who have inspired your work. [...]

- Show curiosity about other people's work. [...]

- Sign up for student volunteer [...]

- Ask your advisor to introduce you [...]

- At the same time, try to socialize outside of your lab [...]

Ultimately, it is important to remember that every first-conference attendee is probably feeling a bit shy too, and that taking the first step in starting a conversation will almost always be met with a big smile!

**[Manually Authored – Writing]**

**I am planning to start a book club with some friends. Can you write an email invitation for the same?**

Here is an example email invitation for a book club:

Subject: Join me for a book club!

Hi [Friend's Name], how are you?

I'm starting a book club with some friends, and I'd love to have you join us!

The book club is a casual and fun way to read and discuss books that we love or want to learn more about, and also make new friends. We plan to meet more-or-less once a month.

We're thinking of having our first meeting on [Date & Time]. Could that work for you?

Would love to see you there!

Thanks,
[Your Name]

---

Figure 10: **Training examples from different sources.** *Top row:* examples mined from community Q&A. *Bottom row:* manually-authored examples. The blue text is the prompt. The gray ellipsis [...] indicates that the response was trimmed to fit this page, but the actual training example is longer.

After training with these six additional examples, we test the model on a few questions with format constraints and observe that LIMA responses greatly improve. We present two examples in Figure 13, from which we can see that LIMA fails to generate proper answers without structure-oriented training examples (left column), but it can generate remarkably complex responses such as a marketing plan even though we do not have any marketing plan examples in our data (right column).

**Imagine that you have a super-intelligent AI assistant, and that you require help with the following question. Which answer best satisfies your needs?**

**Question:** <QUESTION>

**Answer A:**                                                    **Answer B:**

<ANSWER A>                                                       <ANSWER B>

**Comparing these two answers, which answer is better?**
- ■ Answer A is significantly better.
- ■ Answer B is significantly better.
- ■ Neither is significantly better.

Figure 11: Human annotation interface.

You are evaluating a response that has been submitted for a particular task, using a specific set of standards. Below is the data:

[BEGIN DATA]
***
[Task]: {task}
***
[Submission]: {submission}
***
[Criterion]: helpfulness:
"1": "Not helpful - The generated text is completely irrelevant, unclear, or incomplete. It does not provide any useful information to the user."
"2": "Somewhat helpful - The generated text has some relevance to the user's question, but it may be unclear or incomplete. It provides only partial information, or the information provided may not be useful for the user's needs."
"3": "Moderately helpful - The generated text is relevant to the user's question, and it provides a clear and complete answer. However, it may lack detail or explanation that would be helpful for the user."
"4": "Helpful - The generated text is quite relevant to the user's question, and it provides a clear, complete, and detailed answer. It offers additional information or explanations that are useful for the user. However, some of the points of the response are somewhat repetitive or could be combined for greater clarity and concision"
"5": "Very helpful - The generated text is highly relevant to the user's question, and it provides a clear, complete, and detailed answer. It offers additional information, explanations, or analogies that are not only useful but also insightful and valuable to the user. However, the structured of the response is not well-organized and there is no clear progression or logical sequence of different points in the response."
"6": "Highly helpful - The generated text provides a clear, complete, and detailed answer. It offers additional information or explanations that are not only useful but also insightful and valuable to the user. The response is also in a logical and easy-to-follow manner by explicitly using headings, bullet points, or numbered lists to break up the information and make it easier to read."
***
[END DATA]

Does the submission meet the criterion? First, write out in a step by step manner your reasoning about the criterion to be sure that your conclusion is correct. Avoid simply stating the correct answers at the outset. Then print the choice only from "1, 2, 3, 4, 5, 6" (without quotes or punctuation) on its own line corresponding to the correct answer. At the end, repeat just the selected choice again by itself on a new line.

Figure 12: Prompt for ChatGPT evaluation with a 6-scale Likert score. The placeholders "task" and "submission" will be replaced by specific details from the actual case being evaluated.

| LIMA w/o 6 Format Constraint Examples (994 Examples) | LIMA (1,000 Examples) |
| --- | --- |

**[Marketing Plan (Out of Distribution)]**
I want to start a local coffee shop. Create a marketing plan with the following elements: Marketing Goals and Objectives, Define Target Audience, Research Marketing Tactics, Plan Marketing Tactics and Develop Your Timeline and Budget.

---

Coffee Shop Marketing Plan

Executive Summary

The following marketing plan for Java Jolt Coffee Bar outlines an aggressive approach to establishing a thriving coffee shop business in downtown Pittsburgh. Java Jolt will capitalize on the popularity of specialty coffees and the growing demand for WiFi hot spots by offering a comfortable environment where customers can enjoy premium coffee drinks while taking advantage of free Internet access.

Market research indicates that there is a significant opportunity for a new coffee bar located in the heart of the downtown business district. Market trends suggest that consumers are willing to pay a higher price for quality specialty coffees served in a welcoming atmosphere. Additionally, many office workers and students living in the area do not have easy access to high-speed Internet service, presenting an additional revenue stream for the coffee bar.

This marketing plan includes the following elements:

- Market Research supporting the need for a new coffee shop in the target location
- Detailed Description of the products and services offered by Java Jolt
- Target Market Analysis identifying the primary customer segments for Java Jolt
- Marketing Strategy including pricing decisions, promotional activities, and ways to differentiate Java Jolt from competitors
- Implementation Plan detailing the actions necessary to launch Java Jolt successfully
- Budget allocating funds to each component of the marketing plan

A comprehensive marketing plan has been developed based on extensive research into the local market conditions, consumer trends, and competitive landscape. With its unique combination of premium coffee drinks and wireless Internet access, Java Jolt will quickly become a favorite destination for people seeking a relaxing environment to socialize, study, or conduct business.

Here's a marketing plan for your local coffee shop.
**Marketing Goals and Objectives**

- Increase awareness of our brand among consumers [...]
- Drive foot traffic [...] during slow business hours [...]
- Build relationships with influencers in our community who [...]

**Define Target Audience**
Our target audiences include:

- Residents of the neighborhood surrounding our coffee shop
- Office workers near our coffee shop
- Students at nearby schools

We will focus our marketing efforts on these audiences because they are most likely to become regular customers at our [...]
**Research Marketing Tactics**
Here are some marketing tactics that we could use to reach to reach our target audiences and achieve our goals:

- Host events at our coffee shop that appeal to our target audiences (e.g., open mic nights for students or networking events [...]
- Offer discounts during slow business hours to entice [...]
- [...]

**Plan Marketing Tactics**
Here's how we will implement the marketing tactics above:

- Events: We will host one event per month at our coffee shop. For each event, we will send out email invitations [...]
- Discounts: Every Monday through Thursday from 3pm–6pm, we will offer $1 off any drink purchase. To promote [...]
- Local partnerships: We will reach out to five local businesses and schools to ask if we can leave stacks of coupons [...]
- Social media: [...]
- Reviews: [...]

**Develop Your Timeline and Budget**
This is what our marketing timeline looks like:

- Month 1: Launch email newsletter and social media accounts
- Month 2: Start distributing coupons at [...]
- [...]

Here's the budget for each marketing activity:

- Email newsletter: $50/month for MailChimp subscription
- Social media: $100/month for paid ads [...]
- [...]

---

**[Summarization into Bullet Points (In Distribution)]**
Can you summarize this message into bullet points? Gradually slowing job gains and a growing labor force in March [...] Mr. Biden said [...]. More Americans joined the labor force, [...] this spring that he will seek re-election.

---

Though the unemployment rate fell to its lowest level since May 1970, at 4.8 percent, Mr. Biden still faces an economy that leaves many Americans feeling left behind. Many working-class [...]
Mr. Biden's challenge will be persuading those voters that the economic picture has [...] His opponents have already begun painting a bleak portrait of the economy, arguing that most Americans have not benefited from the president's policies.
"The March jobs report shows continued solid job growth but also indicates that we must do better," said Representative Nancy Pelosi of California, the Democratic leader in the House. [...]

Here are the main points from the text:

- Job creation slowed to 236,000 for the month of March, which is close to the level needed to stabilize the economy and prices.
- More Americans joined the labor force, and wage gains fell slightly, both of which should help cool inflation.
- The report underscores the political tensions facing President Biden ahead of his expected announcement that he will seek re-election.

---

Figure 13: Model outputs from test prompts that ask the model to generate according to specified structures. The gray ellipsis [...] indicates that the response was trimmed to fit this page, but the generated text is actually longer.

