# OpenReview forum: "LIMA: Less Is More for Alignment"
_NeurIPS.cc/2023/Conference — NeurIPS 2023 poster_

### Official Review · Reviewer_fPTm · 2023-07-02

**Soundness:** 2 fair
**Presentation:** 4 excellent
**Contribution:** 3 good
**Rating:** 5
**Confidence:** 4

**Summary:**

In this paper, the authors present an LLM finetuned from 65 LLaMa with very limited 1000 samples. LIMA achieves very impressive results in the human evaluation, with competitive results against GPT4/Bard/davinci003. According to my understanding, the main contribution of this paper lies on two aspects: 1. verifying fact that carefully constructed samples enhance LLM more; 2. deriving a well-constructed SFT dataset as well as a strong LLM.

**Strengths:**

1. The paper verifies a fact that carefully constructed samples enhances LLM more
2. The presented model is a very strong competitor to existing non open-source LLM
3. The extensive analysis on data diversity, quality and quantity is useful.

**Weaknesses:**

1. Quality experiments are limited, where its empirical results are mostly from human evaluation on self-constructed dataset;
2. The data construction process is hard to scale
3. The evaluation is conducted on very limited data (~300)

**Questions:**

The claim of this paper is very strong and impressive. I think such a strong claim should be supported by aa more thorough experiments.
1. The test set is from pushshift r/AskReddut and authors' questions. I would be helpful to see its performance under a finegrained evaluation on its capacability, such as knowledge QA (literature, science), reasoning, toxicity, coding and etc.
2. Human evaluation is subjective. I wonder if the authors can provide results with objective metrics.
3. Out-of-istriution analysis is interesting. Is it possible to evaluate LIMA on questions from a third party (e.g. questions from sharegpt)?
4. LIMA is tuned with 1000 samples. Do you try other number of sampels? I wonder how LLM performance changes against increasing quantity of well curated samples. How do you determine the number of well curated samples set for a certain model? And is the conclusion consistent across base models?

If my concerns are addressed, I am happy to raise my score.

**Limitations:**

Yes

---

> ### Author Rebuttal · Authors · 2023-08-10
>
> Question 1: While collecting prompts, Group B assigned each prompt in the test set to a category based on their own definitions. Here are the categories and distribution of the prompts in the test set:
>
> - writing: 13.7%
> - coding and stem: 9%
> - knowledge: 18.7%
> - advice: 18.7%
> - commonsense: 2.7%
> - safety: 2.4%
> - brainstorm: 3.3%
> - roleplay: 3.3%
> - fermi: 1.3%
> - counterfactual: 1.7%
> - constrained generation: 1.7%
> - opinions: 4%
> - NLP: 0.3%
>
> As can be seen, our test set is diverse in terms of tasks. We can add a fine-grained analysis of LIMA’s performance in the appendix with respect to each category in the final version.
>
> Question 2: Please check our general responses. TL;DR: human evaluation is still the least-worst option, given that we do not have a better way to evaluate open-ended long-form generation.
> To the point of subjectiveness: we would also like to point out that reviewing NeurIPS papers is a subjective process, with proven inherent limitations, yet we still do it because we lack a better alternative.
>
> Question 3: The test set does include 70 prompts from questions on AskReddit, which is completely out-of-distribution. The model has never seen data from AskReddit.
>
> Question 4: Our ablations do explore what happens when the model is tuned on more data from the same distribution, demonstrating a “glass ceiling”. However, the final dataset used in LIMA is more diverse, in particular due to the manually-authored responses. Unfortunately, these are expensive to scale, but the contribution of this data is significant. We do hypothesize that alignment tuning can be scaled, but that the scaling law is a function of the dataset’s *diversity* and not only its *quantity*.

---

> > ### Comment · Reviewer_fPTm · 2023-08-19
> >
> > Thanks for the updates. I will keep my score.

---

### Official Review · Reviewer_w2T8 · 2023-07-04

**Soundness:** 3 good
**Presentation:** 3 good
**Contribution:** 3 good
**Rating:** 6
**Confidence:** 4

**Summary:**

This paper presents a superficial alignment hypothesis. It claims that most knowledge and abilities of large language models (LLMs) are learned in the pre-training stage, and a few datasets are just enough to transform the styles of LLMs for alignment.

To support the argument, the authors of the paper finetune LLaMA on the few curated finetuning dataset only having 1000 instances. The curated dataset consists of community-based Q&A such as Stack Exchange, WikiHow, and Reddit, and manually authored examples.

Experimental results (human evaluation on 300 test samples) show that the resulting model, LIMA outperforms baselines including Alpaca 65B, Davinci003, and Bard, and shows compatible performances with other proprietary LLMs, Claude and GPT-4.

Further analyses confirm the results qualitatively, showing generalizability on the safety or multi-turn scenarios. Moreover, the data quality and diversity are suggested to be the most crucial factor for the alignment.

**Strengths:**

1. The hypothesis is convincing, and the experimental results support it well.
2. The main message is clearly delivered.
3. Evaluations and corresponding cross-agreements are well conducted.

**Weaknesses:**

Overall the evaluation is limited.
  - a. More details for the manually designed test prompts are needed, e.g., category, distribution, etc.
  - b. Human (GPT-4) judgment does not guarantee the intended behaviors of the models, according to [1].
    - b-1. There are automatic evaluation benchmarks related to alignment learning, such as HHH evaluation in Big-Bench, TruthfulQA, and RealToxicityPrompt.
    - b-2. Also conventional NLP tasks can also be considered as the target of measurements. Specifically, knowledge probing (MMLU or Reasoning tasks) is a critical factor in getting the model, not just mimicking the preferred styles.
  - The number of examples for the safety or multi-turn analysis is too small. Therefore, I think the claim for the multi-turn dialogue is not convincing, and the analysis should be based on at least 50-100 examples.
  - Ablation study is conducted on the 7B model, unlike the main experiments based on the 65B model. We could not guarantee the same tendency would be observed with the different-sized backbone.

[1] The False Promise of Imitating Proprietary LLMs (Gudibande et al.)

**Questions:**

1. The superficial alignment hypothesis is still valid in the small-sized models? Although you reported the ablation study based on the LLaMA-7B, we do not know that the model generates better responses compared to other baselines.

**Limitations:**

- It would be good to include evaluation results on other automatic evaluation benchmarks (I suggested in the Weakness).
- More details for the manually designed prompts (for the test) make the paper more concrete.

---

> ### Author Rebuttal · Authors · 2023-08-10
>
> Weakness A. While collecting prompts, Group B assigned each prompt in the test set to a category based on their own definitions. This is a noisy process, and the categories are not entirely well-defined, which is why we did not share this information in the paper. However, per your request, we share these categories and the distribution of prompts here:
>
> - writing: 13.7%
> - coding and stem: 9%
> - knowledge: 18.7%
> - advice: 18.7%
> - commonsense: 2.7%
> - safety: 2.4%
> - brainstorm: 3.3%
> - roleplay: 3.3%
> - fermi: 1.3%
> - counterfactual: 1.7%
> - constrained generation: 1.7%
> - opinions: 4%
> - NLP: 0.3%
>
> Weakness B. Thanks for your questions! Please check our general responses.
>
> Weakness C. It is true that the evaluation is based on a limited number of dialogue turns. However, we still find it shocking that LIMA, which was not tuned on multi-turn dialogue at all, is able to conduct some degree of dialogue. We thus feel that it is important to report this observation, even if its evaluation is limited.
>
> Weakness D. We performed ablation studies on the 30B model too (not reported) and we made similar observations as in the 7B model. We can include this in the final version’s appendix.
>
> Q1: The Superficial Alignment Hypothesis explicitly states that the knowledge and capabilities are determined by the pretrained model. Therefore, a weaker (smaller) model should be less capable, even after fine-tuning. Indeed, we observe that while smaller models were able to learn the assistant style from 1,000 examples, the content produced by these models was sometimes lower. For example, in coding prompts, we found that the 7B model was more likely to produce bugs. However, for many other prompts, such as email writing, the differences were not always discernible. These observations are based on our manual analysis of the models; we did not conduct standardized evaluation for 7B models beyond what is presented in the ablations.

---

> > ### Comment · Reviewer_w2T8 · 2023-08-19
> >
> > Thank you for your responses, especially for the prompts category. I raise my score accordingly.

---

### Official Review · Reviewer_xtqr · 2023-07-06

**Soundness:** 4 excellent
**Presentation:** 4 excellent
**Contribution:** 4 excellent
**Rating:** 5
**Confidence:** 5

**Summary:**

The authors investigate on instruction tuning an LLM on high-quality manually collected/created data (only 1K training examples). They provide detailed explanation on how to collect the data, tune a 65B LLaMA model and compare it with other commercial/open-source LLMs. The main experiment results show that the LIMA 65B LLaMA model is better than 65B Alpaca and slightly worse than the top commercial models (GPT, Claude and Bard), based on human and GPT-4 judgements. The authors also conduct other ablation studies including dialogue modeling ability, safety, data quality v.s. quantity.

**Strengths:**

1. The presented data quality is very high. There are many recent papers about instruction tuning on a large amount of instruction data (typically by prompting GPTs), while this paper demonstrates that quality can play a bigger role. This finding is very inspiring for the community.
2. The LIMA model can also somehow reject unsafe queries, which is difficult for Alpaca, Vicuna or other fine-tuned LLaMAs on GPT generated data. This result is surprising to the reviewer as the ability of rejection to unsafe query is more likely to gain from RLHF.
3. The model also show promising performance on dialogue tasks. This is good but actually expected, as Vicuna showed that dialogue ability is not hard to gain.

**Weaknesses:**

1. The test set is kind of small. In addition, although the authors mention they put effort to make the test data diverse and distinct from the training set, the diversity of the test set may still be a issue since all the authors are likely to fall within a narrow scope (i.e., well-trained machine learning researchers). It would be better to include more testing prompts from different people, e.g., real user inputs. This might be much harder for academia folks though comparing with OpenAI and Google.
2. As pointed out in [Gudibande et al.](https://arxiv.org/pdf/2305.15717.pdf), the SFT tuned LLaMA models can receive good ratings comparable to ChatGPT from crowd workers but actually have a huge gap, and they called it as a false promise. The evaluation of the LIMA model is very limited to human/GPT judgments. Whether it has the similar issue is unknown from the current presentation. Maybe it's better than LLaMAs tuned on GPT generated data; or it also has the same issue --- which may indicate that RLHF's importance. Current evaluation cannot provide the readers enough information for this.
3. Many recent papers have shown that the real emergent ability, aka what makes GPTs/Bard/Claude different from the others are complex task solving. See [Yao et al.](https://yaofu.notion.site/Towards-Complex-Reasoning-the-Polaris-of-Large-Language-Models-c2b4a51355b44764975f88e6a42d4e75), and also OpenAI’s [GPT-4 release blog](https://openai.com/research/gpt-4): “In a casual conversation, the distinction between GPT-3.5 and GPT-4 can be subtle. The difference comes out when the complexity of the task reaches a sufficient threshold”. The reviewer would like to see the analysis on these kind of tasks (e.g., MATH/BB-Hard) as well.

**Questions:**

In line 78, "Within each exchange, we take the questions with the highest score that are self-contained in the title (no body)". The reviewer doesn't understand this sentence. Could you clarify the meaning of the terms here? What is the score for, what is "no body" and what does self-contained mean here?

**Limitations:**

The way of manually collecting high quality data could be expensive and hard to scale. It's still a question that given the same amount of money, which option is more effective: manually collect data and guarantee quality; or just prompt powerful closed-source LLMs to get a huge amount of data.

---

> ### Author Rebuttal · Authors · 2023-08-10
>
> Weakness 1: The test set does indeed include naturally occurring prompts from Reddit.
>
> Weakness 2: We agree that there are limitations to evaluating models by human preferences. However, there are even more significant limitations to evaluating models by other existing benchmarks, e.g. MMLU, which are completely misaligned with the actual usage of assistants/chatbots. Please provide a constructive suggestion for evaluating this type of functionality.
>
> Weakness 3: Indeed, but these are all examples of advances in pretraining, which is not in the scope of this paper. LIMA focuses on showing that the stages following pretraining can be greatly simplified.
>
> Q1: In Stack Exchange, each question is scored (up/down-voted) by the users. We select the top questions. In addition, each question has a “title” and a “body” (detailed description); we only use the self-contained title as our prompt without the “body”.

---

> > ### Comment · Reviewer_xtqr · 2023-08-10
> > **Replies and Further Questions**
> >
> > > Weakness 1: The test set does indeed include naturally occurring prompts from Reddit.
> >
> > Given the portion of examples from reddit (~20+%) and from authors (70 from reddit v.s. 230 from authors), the concerns in the original review still stand. Reporting a separate number for the reddit subset can be the simplest way to address this.
> >
> > > Weakness 2: We agree that there are limitations to evaluating models by human preferences. However, there are even more significant limitations to evaluating models by other existing benchmarks, e.g. MMLU, which are completely misaligned with the actual usage of assistants/chatbots. Please provide a constructive suggestion for evaluating this type of functionality.
> >
> > There are many ways to do so in both academia and industry. For example, the HHH principle (Helpfulness, Faithfulness, Harmless). Different aspects of the result quality, e.g., task following, creativity, fluency etc. A simple "model x beating y" means little in the actual use case and is far from enough. There are many papers discussed these topics and benchmarks available, e.g., see Anthropic's paper series.
> >
> > > Weakness 3: Indeed, but these are all examples of advances in pretraining, which is not in the scope of this paper. LIMA focuses on showing that the stages following pretraining can be greatly simplified.
> >
> > The emergent ability phenomenon also exists in instruction tuning. See Sec 4 in [Wei et al.](https://arxiv.org/pdf/2206.07682.pdf). I may provide wrong example tasks though.
> >
> > > The way of manually collecting high quality data could be expensive and hard to scale. It's still a question that given the same amount of money, which option is more effective: manually collect data and guarantee quality; or just prompt powerful closed-source LLMs to get a huge amount of data.
> >
> > No comments addressing this yet. Accessing human labels are expensive, especially when requiring high quality annotations. Given the same amount of money, it's actually possible that calling APIs can generate an instruction tuning dataset with lower quality but still achieves similar outcome.

---

> > > ### Author Response · Authors · 2023-08-10
> > >
> > > No problem! We actually wrote this for reviewer a3cj, but here is the result table copied over:
> > >
> > > |                   | Pushshift r/AskReddit (70) | Authors (230) | ALL (300)      |
> > > |-------------------|-----------------------------|---------------|----------------|
> > > | LIMA vs Alpaca (65B)   | 64%/16%/20%               | 50%/22%/28%   | 53%/21%/26%    |
> > > | LIMA vs DaVinci003     | 57%/13%/30%               | 40%/23%/36%   | 44%/21%/35%    |
> > > | LIMA vs BARD (April)   | 30%/29%/41%               | 33%/24%/43%   | 33%/25%/42%    |
> > > | LIMA vs Claude (April) | 20%/27%/53%               | 25%/20%/55%   | 24%/22%/54%    |
> > > | LIMA vs GPT4           | 22%/17%/60%               | 17%/27%/56%   | 18%/25%/57%    |
> > >
> > > As we can see from the table, LIMA’s performance on AskReddit prompts is indeed similar to its performance on Group B’s prompts. In fact, when compared to Alpaca or DaVinci003, performance on AskReddit is significantly better.
> > >
> > >
> > > Alternative Evaluations: The HHH principle promoted by Anthropic is interesting, and certainly has merits, but it is only way of defining "aspects" or "dimensions" by which to compare models. It is not necessarily the ground truth.
> > >
> > >
> > > Scaling Data: Actually, distilling data from APIs is scientifically problematic - it assumes that a "gold" model already exists. We are interested in finding an efficient recipe for creating aligned language models without dependency on other aligned models.

---

### Official Review · Reviewer_s3dY · 2023-07-07

**Soundness:** 3 good
**Presentation:** 2 fair
**Contribution:** 3 good
**Rating:** 7
**Confidence:** 3

**Summary:**

This paper presents LIMA, a 65 billion-parameter large language model fine-tuned on a dataset of 1,000 carefully curated prompts and responses without the use of reinforcement learning or human preference modeling. The aim of this research was to measure the relative importance of pretraining and large-scale instruction tuning in the training of language models. LIMA demonstrates strong performance and generalizability across a variety of tasks, even those not included in its training data. It also excels in specific response formats, learning from just a few examples in the training data. The study includes a controlled human evaluation where LIMA's responses were either equivalent or preferred to those of GPT-4 in 43% of cases. When compared to Bard and DaVinci003, this percentage rose to 58% and 65%, respectively. The results suggest that the majority of knowledge in large language models is acquired during the pretraining stage, and only limited instruction tuning data is required to generate high-quality output. Ablation experiments underscored the diminishing returns from scaling data quantity without a concurrent increase in prompt diversity, highlighting the critical role of data quality and diversity in training these models.

**Strengths:**

1. The paper presents an effective approach to training large language models, focusing on the importance of pretraining and using a limited number of carefully curated prompts for fine-tuning. This approach challenges the common practice of using large-scale instruction tuning and reinforcement learning.
2. The experimental results indicate the effectiveness of the proposed training approach, which shows strong performance compared to various LLMs, including commercial ones.
3. Detailed ablation studies are given regarding the data diversity, quality, and quantity, which allows the readers to better understand different aspects of the proposed approach (and data).

**Weaknesses:**

1. The proposed training scheme is only tested with subjective evaluations. I wonder if the proposed method can also give significant improvements on objective benchmarks, such as MMLU, etc.
2. Also, the evaluation can be further improved by introducing different settings, such as CoT, few-shot, etc.
3. It is not quite clear how the conclusions can be applied to other scenarios, such as different languages, domains, etc. The paper can be further strengthened by providing some takeaways to the readers.

**Questions:**

Question:

1. Please address the weaknesses above.
2. Will the 1,000 examples and LIMA model be released to the public to further promote open research?
3. As the training process only involves 1K examples, why the experiments are only performed on 65B LLaMA? I know that the ablations are carried out on 7B model, but it would be also interesting to see a progressive analysis when model scales from 7B, 13B, 33B to 65B.



Other comments: There are some typos and grammatical issues that should be corrected.

1. line 2: large scale -> large-scale
2. line 37: LLaMa -> LLaMA
3. line 163: state of the art -> state-of-the-art
4. line 206: Of -> of


**Limitations:**

The authors discussed their limitations in Section 7.

---

> ### Author Rebuttal · Authors · 2023-08-10
>
> - Weakness1: We intentionally did not evaluate using MMLU and other multiple-choice or classification benchmarks since they are unrelated to the functionality that we tried to imbue in the model. The experiments in LIMA show that an assistant (chatbot) can be trained from a strong pretrained model using only 1,000 high-quality examples. We were therefore interested in evaluating LIMA on prompts that are likely to be asked by assistant users, e.g. life advice, creative writing, coding, etc.
>
> - Weakness 2: Again, both CoT and in-context learning are not aligned with how laypeople are likely to use a chatbot, and it therefore does not make sense to evaluate it in those settings. We expect the model to produce high-quality outputs directly, in a zero-shot setting, without any assistance from the prompt.
>
> - Weakness 3: The same methodology can be applied to any scenario, given a strong enough pretrained model and a set of high-quality examples in the style of the desired functionality.
>
> - Q2: The examples, including the test set, are available online.
>
> - Q3: We did experiment with smaller models (see ablations), though we do not report human evaluation compared to other SotA models for those. We can add GPT-4 based evaluation to the paper’s appendix.

---

> > ### Comment · Reviewer_s3dY · 2023-08-21
> >
> > Thanks for your response. I'll keep my original rating.

---

### Official Review · Reviewer_a3cj · 2023-07-20

**Soundness:** 3 good
**Presentation:** 4 excellent
**Contribution:** 3 good
**Rating:** 7
**Confidence:** 4

**Summary:**

This paper hypothesizes that alignment of pretrained LLMs to become user friendly and useful AI chatbots is mainly about learning the style or format of interacting with users, and that all/most knowledge and capabilities were already acquired by the LLM during pretraining. To test this hypothesis, this paper curates 1,000 diverse high quality examples that approximate real user prompts  along with high-quality responses, and trains a new model called LIMA. LIMA demonstrates good performance in a controlled human study; responses from LIMA are either equivalent or strictly  preferred to GPT-4 in 43% of cases; this statistic is as high as 58% when compared to Bard and 65% versus DaVinci003, which was trained with human feedback.


**Strengths:**

This paper addresses an important and timely question aiming to shed light on the mechanisms behind the huge success of instruction tuned LLMs.  The paper is very well written and easy to follow. I really like the straightforward approach of the paper: it clearly defines the crisp question regarding the role of instruction tuning, and opts to answer it directly by creating a small, clean, and diverse training set. The resulting model is clearly a very good instruct model, and achieving this so simply is a very impressive feat. Judging this paper from an academic standpoint, I do have some concerns, detailed below.



**Weaknesses:**

If I understand correctly, table 1 implies that over 75% of the test set is written by the paper authors (group B) which received the same general instructions as the paper authors who wrote 20% of the training set and the entire dev set questions (group A). I see this as weakening the paper’s messages, in particular the comparisons to other models in figures 1 and 2 could be unclean.

One way to put it, as I think the paper is trying to do, is that group B are different people than group A with different interests, friends, etc, and therefore the test questions of group B can be considered an unrelated ground truth test. Another way to put it, which I think is valid, is that LIMA enjoys a significant amount of training on data that is qualitatively closer to the test distribution than any of the data used for training any of the other models. This is concerning since LIMA can be said to enjoy significant training on the tets distribution while other models don’t; this is known to have dramatic effects.

An ablation that slices the results of figure 1 according to 230 test examples written by group B and 70 test examples from Pushshift r/AskReddit can ease my concern here; if the two groups exhibit similar trends, then training on data from group A did not bias the results. If performance on the  230 test examples written by group B is way better, then I think the messages need to be revised. Another experiment that could ease my concern is training “clean-LIMA”  without the data from group A, and supplementing it with 200 examples from the other sources of the training set. A third alternative, perhaps better than the former, is to devise a "clean test set" without examples written by group B. I realize that the last two require further annotation and may be harder, but the authors did show that automatic evaluation with GPT-4 correlates with human judgement and this can be a place to leverage that.

If the authors convince me that I either misunderstood their setup or that training from group A did not bias the test results,I’d be happy to raise my soundness score and overall evaluation.


**Questions:**

How do you fine-tune Llama on the Alpaca dataset?  Is there a standard setting which the authors released? Did you perform a hyper-param search? Since this is the only baseline model that LIMA beats, and the only one that you trained in-house, it is important to make sure that it receives at least equal attention in training as your model. I’d be happy if you could elaborate exactly on how you trained both LIMA and your version of Alpaca.

**Limitations:**

Yes

---

> ### Author Rebuttal · Authors · 2023-08-10
>
> Thank you for pointing out this potential issue!
>
> We acknowledge the reviewer’s concern regarding potential similarities between Group A and Group B data, and provide the test set results (LIMA win / tie / LIMA lose), split 230 / 70 by prompt source, demonstrating that the trends are not over-specific to underlying similarities between Group A and Group B:
>
> |                   | Pushshift r/AskReddit (70) | Authors (230) | ALL (300)      |
> |-------------------|-----------------------------|---------------|----------------|
> | LIMA vs Alpaca (65B)   | 64%/16%/20%               | 50%/22%/28%   | 53%/21%/26%    |
> | LIMA vs DaVinci003     | 57%/13%/30%               | 40%/23%/36%   | 44%/21%/35%    |
> | LIMA vs BARD (April)   | 30%/29%/41%               | 33%/24%/43%   | 33%/25%/42%    |
> | LIMA vs Claude (April) | 20%/27%/53%               | 25%/20%/55%   | 24%/22%/54%    |
> | LIMA vs GPT4           | 22%/17%/60%               | 17%/27%/56%   | 18%/25%/57%    |
>
> As we can see from the table, LIMA’s performance on AskReddit prompts is indeed similar to its performance on Group B’s prompts. In fact, when compared to Alpaca or DaVinci003, performance on AskReddit is significantly better.
>
> We would also like to emphasize that closed models (DaVinci003, GPT-4, Bard, Claude) were possibly exposed to similar data during their training; much of the data in both Group A and Group B is inspired by how people use ChatGPT.
>
> Comparison to Alpaca: We follow the training protocol of Alpaca by training for 3 epochs and other regularization techniques we used for LIMA training. We also performed hyperparameter search over training steps and regularization strengths but didn’t find any significant differences.
> We also emphasize that LIMA also outperforms DaVinci003, not only Alpaca, though to a lesser extent.

---

> > ### Comment · Reviewer_a3cj · 2023-08-13
> > **Raising my score**
> >
> > I thank the authors for clarifying the issue. Indeed, it looks like the similarity between groups A and B did not bias the results. I am raising my score accordingly.

---

### Author Rebuttal · Authors · 2023-08-10

## General Response

A general concern among reviewers is that the human evaluation in LIMA is limited, and that the paper should also include automatic benchmark evaluations. While we agree that there are limitations to evaluating models by human preferences, we argue that there are even more significant limitations to evaluating **generative** AI models via existing benchmarks, especially when considering long-form responses to open-ended prompts. This problem stems from the fact that conversational prompts can have multiple good, correct responses that are completely different from one another. For example, if we ask a model to generate a humorous poem about the reviewing process at NeurIPS, one can imagine many excellent responses that wildly differ in style, content, and lexical choice. Therefore, human evaluation, with all its limitations, is the “least worst” option for evaluating the generative and conversational capabilities we hoped to imbue in LIMA.

In contrast, the proposed benchmarks, e.g. MMLU, focus on testing knowledge. They are completely misaligned with the actual usage of generative AI assistants, in both format (multiple choice vs open-ended generation) and in spirit (concrete knowledge vs creativity). We therefore do not use these benchmarks.

## Response to Ethics Reviewer KQa4
1. As mentioned by the reviewer, the paper acknowledges that any work on generative models includes the risk of generating toxic content. Since the model itself was not released, the risk of laypeople misusing the model is mitigated.

2. The paper does conduct some degree of safety analysis. We emphasize in the paper that the model is not product-grade in terms of safety.

3. Crowd workers were compensated at 15 USD / hour, via Amazon Mechanical Turk, after a training and vetting phase where annotators were also paid at a rate of 15 USD / hour.

## Response to Ethics Reviewer 43fz
Crowd workers were recruited on Amazon Mechanical Turk following standard protocols of paid vetting (15 USD / hour). We do not know and cannot provide any identifying information of crowdworkers, since it is not ethical.

---

### Decision · Program_Chairs · 2023-09-21

**Decision:**

Accept (poster)

**Comment:**

This paper addresss the importance and effectiveness of pretraining and fine-tuning for large language models, which is an important topic.

All reviewers initially gave acceptance scores and some reviewers raised the score during the rebuttal period.

AC also agrees with the reviewers, so decides to recommend accepting this paper.